# Impacts of Climate Change on the Water Resources of the Kunduz River Basin, Afghanistan

**Noor Ahmad Akhundzadah [1],*, Salim Soltani [2] and Valentin Aich [3]**

1    Faculty of Environment, University of Kabul, Kart-e-Sakhi, Kabul 1001, Afghanistan
2    Institute for Geography and Geology, University of Würzburg, Am Hubland, 97074 Würzburg, Germany;
     mohammad_salim.soltani@stud-mail.uni-wuerzburg.de
3    Potsdam Institute for Climate Impact Research (PIK), Am Telegraphenberg, 14473 Potsdam, Germany;
     vaich@wmo.int
*    Correspondence: noorahmad.akhundzadah@ku.edu.af; Tel.: +93-(0)-707083359

**Abstract:**   The Kunduz River is one of the main tributaries of the Amu Darya Basin in North Afghanistan.  Many communities live in the Kunduz River Basin (KRB), and its water resources have been the basis of their livelihoods for many generations.  This study investigates climate change impacts on the KRB catchment.  Rare station data are, for the first time, used to analyze systematic trends in temperature, precipitation, and river discharge over the past few decades, while using Mann–Kendall and Theil–Sen trend statistics.  The trends show that the hydrology of the basin changed significantly over the last decades.  A comparison of landcover data of the river basin from 1992 and 2019 shows significant changes that have additional impact on the basin hydrology, which are used to interpret the trend analysis.  There is considerable uncertainty due to the data scarcity and gaps in the data, but all results indicate a strong tendency towards drier conditions.  An extreme warming trend, partly above 2 °C since the 1960s in combination with a dramatic precipitation decrease by more than −30% lead to a strong decrease in river discharge. The increasing glacier melt compensates the decreases and leads to an increase in runoff only in the highland parts of the upper catchment. The reduction of water availability and the additional stress on the land leads to a strong increase of barren land and a reduction of vegetation cover. The detected trends and changes in the basin hydrology demand an active management of the already scarce water resources in order to sustain water supply for agriculture and ecosystems in the KRB.

**Keywords:** climate change; Kunduz River Basin; trend analysis; river discharge; landcover changes

## 1. Introduction

Afghanistan is a semi-arid country with high variability and irregularity in precipitation.  Based on the morphological and hydrological systems of Afghanistan, its surface water is divided into five major river basins: Kabul, Helmand, Harirud-Murghab, Northern, and Amu-Darya River Basins [1] (Figure 1). The Kunduz river is one of the main tributaries of the Amu Darya in North Afghanistan. It is mainly nourished by snow and glaciers melting during spring and summer (Figure 1).  Similar to other tributaries of the Amu-Darya, it is the main water resource for drinking, irrigation, and hydropower usages in the basin and the river plays an important role for all ecosystems in the basin [2–4].  However, riverine floods and flash floods are common disasters in the Kunduz River Basin (KRB), because of the extreme climate regime in the Hindu Kush Mountains. Severe riverine flooding in the lowlands and upper parts of the catchments occur regularly during spring due to glacier and snow melt and spring rainfall.  In the year 2019, early rainfall in upper parts of the catchments, combined with increased snowmelt due to high temperatures, caused strong flooding in most river basins of the Amu

Darya tributaries in Afghanistan, with approximately 124,500 people affected and many killed [5]. Little literature is available on climate change impacts in Afghanistan; some recently conducted studies indicate a distinct warming trend and a decrease of rainfall in some parts of the country [6,7]. The first detailed and systematic analysis of climate data for Afghanistan that was conducted by Aich et al. (2017) showed a warming by 1.8 °C for Afghanistan between 1951 and 2010; the temperature in Afghanistan increased by 1.8 °C, which is higher than the global mean. These changes severely affected the key sectors, including water resources, agriculture, energy, and it imposed flash flood, drought, soil erosion, and environmental degradation [8–11].

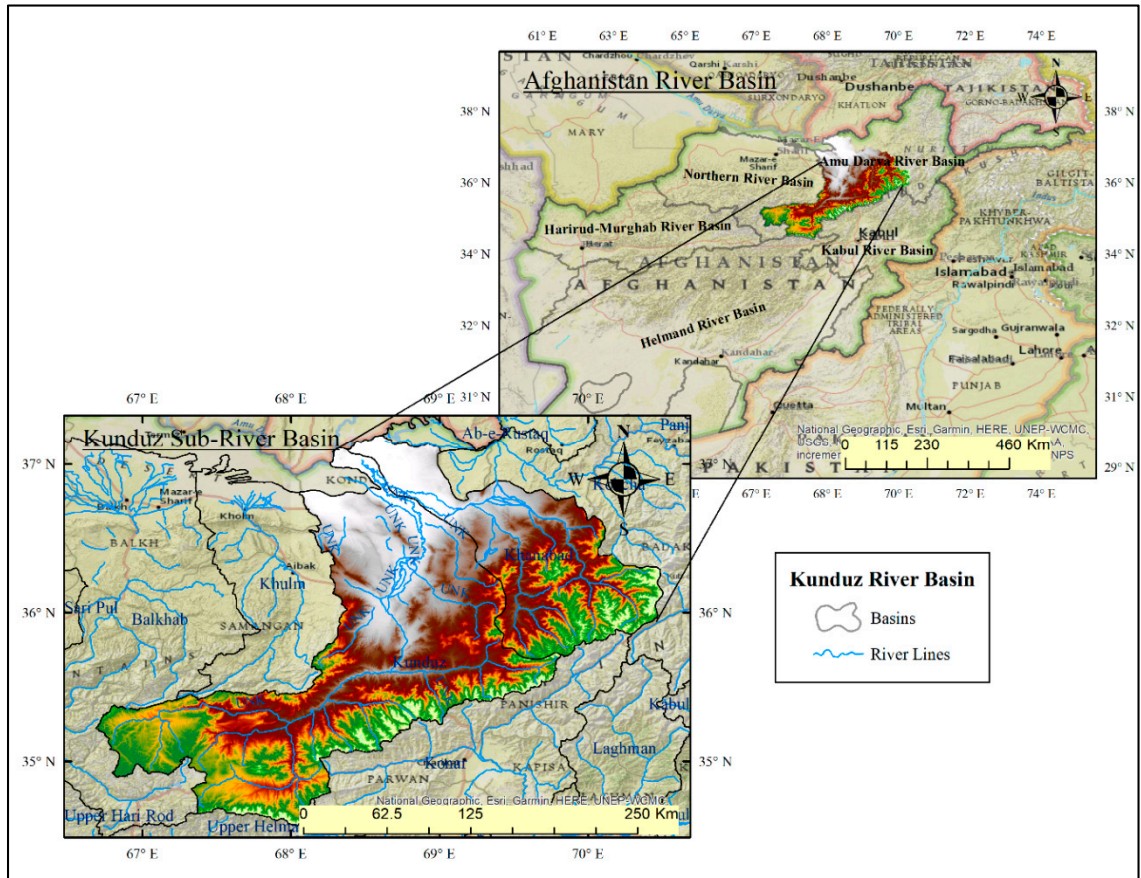

**Figure 1.** Afghanistan main river basins and Kunduz river watershed location.

Because about 80% of Afghanistan's population depends on agriculture for their livelihoods, and agriculture contributes to almost half of the GDP [12], these changes directly affected livelihoods, food security, and the socio-economy of the country [10,13]. The changing climate has changed the hydrological condition and land cover of the Amu-Darya River Basin [11,14]. The increase in temperature has been melting glaciers and permafrost in Himalayan and Hindukush mountains [1,15,16]. The decrease in precipitation and glaciers melting has reduced the volume of water in the Amu-Darya and KRB [2,4,17].

The climate change impacts, compounded by the past four decades of war and conflict, have destroyed the country's infrastructure and institutions, and it has led to underdevelopment that collectively contributes to Afghanistan's vulnerability to climate change impacts. Now, any climate change study in Afghanistan is faced with the challenge of lack of reliable historical meteorological data, with more than two decades gaps in the historical data records during the war and conflict in the country [18]. The related uncertainties are also reflected in global reanalysis products [9].

However, so far, no study has addressed the impacts of these different factors on the water resources of the catchment with all available observed data. Therefore, the main focus of this study is to investigate climate change impacts in the hydrology of the KRB with a focus on temperature, precipitation, river discharge, and land use and land cover (LULC) change, while taking into account the lack of data and the resulting uncertainty. Therefore, the trends of these variables are analyzed and the results integrated in a discussion. The observed data from the KRB are mainly available for the period 1960s–1980s and then again from the 2000s until now with a large gap in between due to the political conflicts in Afghanistan, which hinders the trends analysis. The limitation in data availability and its implications on the study and its results are discussed when interpreting the results. Finally, conclusions for water resource management in the basin are drawn while taking the data constraints and the related uncertainty into account.

## 2. Study Site

### 2.1. Kunduz River Basin

The Kunduz River is one of the main tributaries of the Amu Darya. It originates from the North side of the Hindukush Mountain and flows through the wide lowlands of Baghlan to finally join the main Amu Darya stream in Qala-i-Zal area (Figure 2). The Kunduz watershed has an area of 28,024 km$^2$, which is 4.5% of the country [19] and about 1.9% of the population of the country live in the River Basin [20]. The KRB covers the mountainous area of the Hindukush, with elevation ranging from up to 4000 m a.s.l. in the upper, Southern parts of the Basin. Lowland areas are about 600 m a.s.l. in Baghlan and 400–350 m a.s.l. in Kunduz provinces. The soils of the KRB are characterized by Palaeogene and Neogene sediments and covered by Loess deposits about 30 m to more than 100 m thickness in the center. Alluvial deposits consist of gravel, sands, and silt spread around floodplain in the basin. The area adjacent to the mountains are covered by coarse deposits of gravel, pebble, cobble, and other detritus deposits [21,22]. The higher altitude areas in the basin are partly used for rain-fed agriculture, but they mostly consist of deforested areas [23]. The flood plains consist of highly fertile medium drained soils with good agricultural land, which comprises the main economic center of the basin [24].

Arable land covers 38% (10,344 km$^2$) of the total area of the KRB (28,024 km$^2$). The Takhar province has more arable land as compared to the Kunduz and Baghlan provinces. Bamyan province is located in the high mountain area of the KRB and it has the least arable land [23]. The main crops cultivated in the arable area of the KRB are wheat, maize, barely, and rice. The crops are mostly planted during March to May and harvesting during July to September [25]. Watermelon, melon, potatoes, and onions are the main vegetables crops. Apples, grapes, berries, and peaches are the major fruits, and Cotton is the major industrial crop. There is an increasing number of pistachio- and almond plantations grown in the KRB.

Recently, the Ministry of Agriculture, Irrigation and Livestock (MAIL) conducted qualitative and quantitative investigations on climate change impacts on the agriculture sector in Afghanistan. The results presented a significant reduction in crops production, which was likely due to a decrease of precipitation and rising temperatures within the North-East agro-climatic zone that covered the KRB (e.g., 10–20% reduce in wheat) [25].

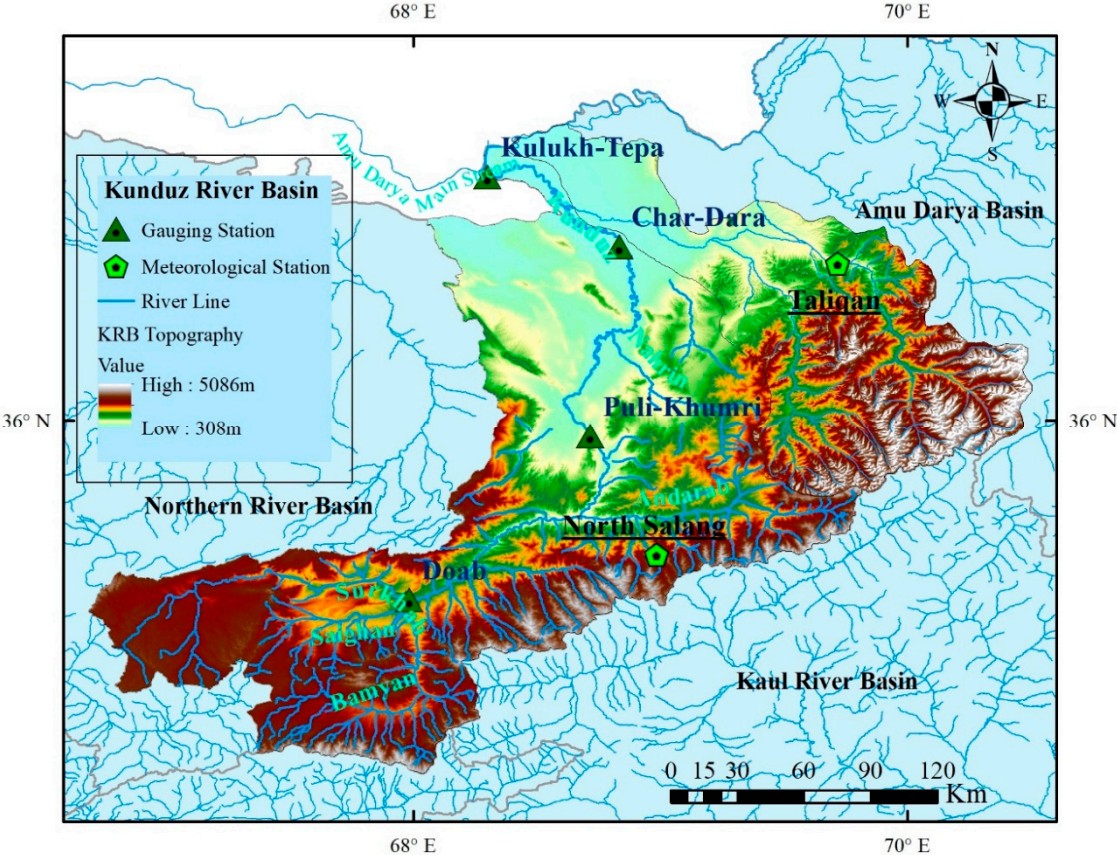

**Figure 2.** Main tributaries and locations of hydrologic stations and stream gauges within the Kunduz River Watershed.

## 2.2. Climate

Precipitation and temperature are very heterogeneous in the KRB due to its large range in elevation. Based on the Köppen–Geiger climate classification scheme, the KRB is mainly characterized by a mid-latitude steppe climate (Bsk, cold semi-arid climate) with some areas being Mediterranean-influenced subarctic climate (Dsc) [25]. Figure 3 presents the mean monthly weather average of the recent decade (2009–2019) mean monthly weather average, recorded in North Salang and Kunduz stations. The data were provided by the Afghanistan Meteorological Department [26]. The mean annual temperature in North Salang (3400 m a.s.l.) is around 1 °C and it is 19 °C in Kunduz (991 m a.s.l.). The mean annual rainfall is recorded 71 mm in North Salang and 32 mm in Kunduz. From June to September are mainly dry months with very little precipitation and most of the annual precipitation falls from January to April. At North Salang, the annual average precipitation is around 200 mm and 100 mm at the Kunduz station. July is the warmest month of the year, in North Salang the average temperature in July is 11 °C and, in Kunduz, it is 33 °C. January is the coldest month of the year, with −10 °C and 5 °C in North Salang and Kunduz, respectively. In Kunduz, the temperature extremes can rise to over 40 °C during the warmest months and fall to −20 °C during the cold season. There are occasions of heavy precipitation events, for example, over 400 mm/d in North Salang (e.g., March of 2019) and 350 mm/d in Kunduz (e.g., February 2008). High precipitation during spring 2019 caused severe flash floods in the main river basins, including the Kunduz sub-river basin [27].

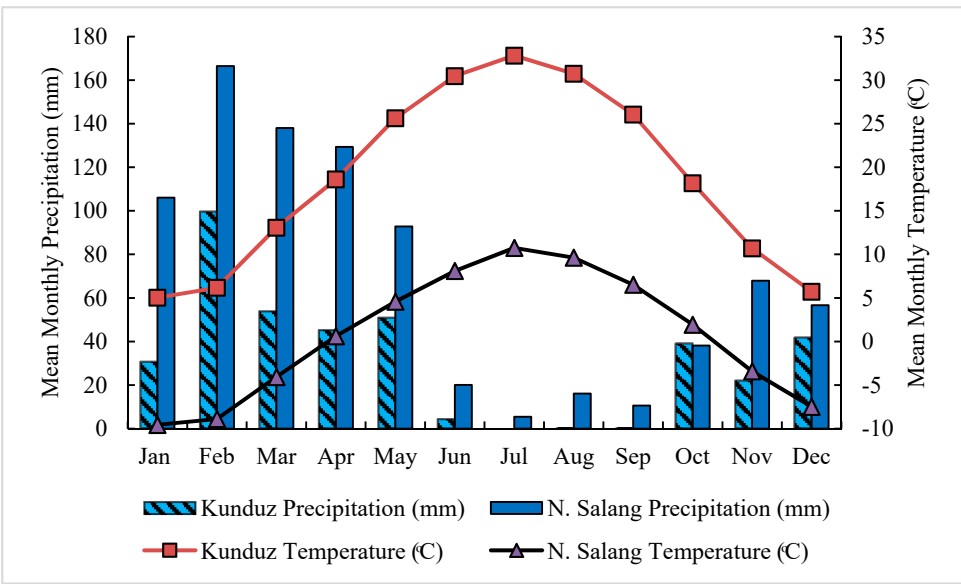

**Figure 3.** Average monthly precipitation and temperature recorded in North Salang and Kunduz stations during the period 2009–2019.

*2.3. Hydrology*

The Kunduz River is a tributary of the Amu Darya River in North Afghanistan. The upper part of the KRB is characterized by high mountains and steep valleys. In the upper part, the KRB is fed by the rainfall, snow, and small glaciers of the Koh-e-Baba range and the Hindu-Kush mountains [24] (Figure 2). The KRB has a number of tributaries, including the Khinjan, Andarab, and Bamyan rivers [28]. Upstream of the Kunduz province, the Kunduz river is called the Pul-I Khumri River. Another small tributary, the Nahrin River, has its sources in Nahrin district and it joins the Kunduz river near the town Baghlan-i Kohna. Finally, the Kunduz River reaches the Amu Darya main stream at Qala-i Zal (Figure 3). The KRB covers all of Baghlan province, the western part of Bamiyan province, and parts of Kunduz and Takhar Provinces [23]. Two hydropower dams have been built on the Pul-i Khumri in 1943 [29].

The hydrology of the KRB is mainly controlled by the high mountains of the Hindukush. Upstream, channels are generally narrow and deep and flowing throughout the whole year [12]. The runoff regimes are largely controlled by snow-melt, with high discharge from April to June and only close to glaciers in the upstream parts of the catchment, the small glaciered area has significant influence on the flow regime (e.g., Doab station). Precipitation in the KRB mainly occurs in the form of rain, drizzle, snowfall, and hail, and it is high during the winter months [24]. The water carried by the river supports an intensive irrigated agriculture, which is the main economic basis of the region. There are a number of river gauging stations within the watershed, as shown in Figure 2.

Figure 4 presents the mean monthly discharge of the recent five years from 2014 to 2018 recorded in the four main gauging stations, Doab, Puli-Khumri, Char-Dara, and Kulukh-Tepa (for locations, see Figure 2). Historically, the monthly peak flows generally occurred during April through July, which resulted in very high discharge at the downstream drainage outlet (Figure 4). The Doab gauge is located in the most upper part at 1468 m a.s.l. It covers a small watershed and has low discharge, being mainly fed by small glaciers. The peak monthly discharge at that gauge from 2014 to 2018 was 36 $m^3$/s during June. The gauge at Puli-Khamri is downstream at 634 m a.s.l. and its peak monthly discharge during this period was 199 $m^3$/s. Char-Dara gauge, further downstream at 401 m a.s.l., the peak discharge is 138 $m^3$/s and 177 $m^3$/s at Kulukh-Tepa gauge (320 m a.s.l.). The Kulukh-Tepa gauging station is located at the confluence of the Kunduz River and the Amu Darya mean stream (Figure 3). The peak average monthly discharge at Puli-Khamri gauge is higher than the Kulukh-Tepa

at the outlet of the KRB. This can be explained by the high temperature and related high evaporation during June, July, and August in the lowland downstream area and diverging small portion of the stream to irrigation as well.

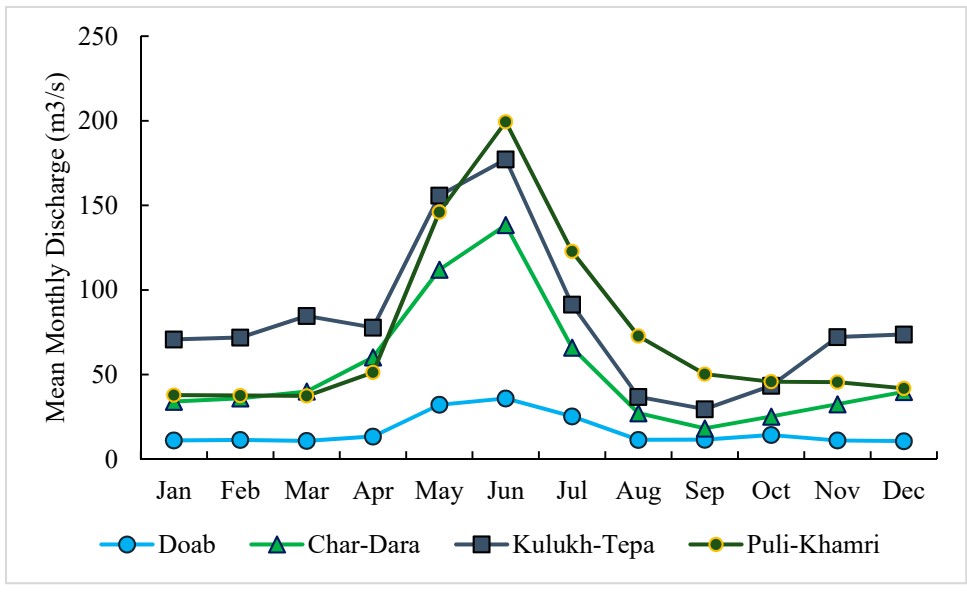

**Figure 4.** Comparison of flow discharge at Doab, Char-Dara, Puli-Khamri, and Kulokh-Tepa gauges.

## 3. Data and Methods

### 3.1. Data

For the analysis, historical temperature, precipitation, and river discharge data recorded from about 1960 to 1979 at gauging stations installed within the Kunduz River Basins are analyzed. Between 1979 and 2009, or even later, there are no data records due to political turmoil in the country available. The recent data from 2009 to 2019 are, with some exceptions, available. The data for available years are listed in Table 1. Only meteorological and river discharge stations with over 20 years of data records have been used in the study and stations with less data available neglected in order to have a minimum of confidence in the time series (discussed in more detail in Section 5.1). The hydrological and meteorological data were provided by the Ministry of Energy and Water and Afghanistan Meteorological Department. The river discharge data are also provided by the same ministry [30].

**Table 1.** Overview of gauges and meteorological stations in the Kunduz River basin, including drainage area, elevation, and the period for which data are available.

| River Gauge Stations | | | | | | | |
|---|---|---|---|---|---|---|---|
| Station Name | Lat. | Long. | Elevation (m) | Drainage Area (km²) | Record Period | Record Period | N° Years |
| Doab | 35.2666667 | 67.9833333 | 1468 | 5005 | 1968–1979 | 2009–2018 | 22 |
| Puli-Khumri | 35.9333333 | 68.7166667 | 639 | 17405 | 1950–1968 | 2009–2018 | 29 |
| Char-Dara | 36.7000000 | 68.8333333 | 401 | 24820 | 1964–1980 | 2007–2018 | 29 |
| Kulokh-Tepa | 36.9833333 | 68.3000000 | 320 | 37100 | 1966–1980 | 2014–2018 | 20 |
| Meteorological Stations | | | | | | | |
| North Salang | 35.4528396 | 68.9852142 | 3400 | Met-Station | 1960–1978 | 2010–2019 | 29 |
| Taliqan | 36.6333333 | 69.7166667 | 991 | 4110 | 1969–1978 | 2010–2019 | 20 |

For topographical and hydrological mapping, remote-sensing data and satellite images from sources, including National Geographic and Esri, were accessed and processed using ArcGIS software (https://www.arcgis.com/home/item.html?id=b9b1b422198944fbbd5250b3241691b6). For the LULC classification, Landsat 5 Thematic Mapper (TM) scenes and Landsat 8 Operational Land Imager (OLI) have been used [31].

## 3.2. Trend Analysis for Temperature, Precipitation and River Discharge

Linear trends in the time series were analyzed using the Mann–Kendall test [32]. It was chosen, because it is a robust nonparametric test and it can handle missing data as well as it has higher power for non-normally distributed data, which are common in hydrological and meteorological data [33]. Each element is compared with its successors and ranked as larger, equal, or smaller. Based on this analysis, the statistical significance of rejecting the null hypothesis that there is no monotonic trend is tested (for all tests $\alpha = 0.05$). The R package "Kendall" was used for the calculation [34].

The Theil–Sen approach was used in order to quantify the linear trend [35,36]. It computes the slope for all pairs of the ordinal time points of a time series and then used the median of these slopes as an estimate of the complete slope. This approach is commonly combined with the Mann–Kendall test and estimates the trend slope of a time series in its original unit. The R package "zyp" was used in order to calculate the Theil–Sen trend and includes a pre-whitening according to Ye et al. (2002) [37] if autocorrelation occurs [38].

## 3.3. Land Cover Classification

The supervised land cover classification has been carried out in two time steps, 1992 and 2019, while using Landsat 5 TM for the earlier date and for the latter Landsat 8 OLI. To account for annual variation in the snow and glacier coverage, data from August and September, when snow and glacier coverage have their annual minimum, have been used. A cloud mask was applied to remove cloud contamination.

The Random Forest Classifier (RFC) method [39] was applied using Google Earth Engine (GEE) for the classification [40]. For constructing the study wide cloud free mosaic, the median function of GEE has been used, which takes the median value of each pixel in available image temporal stack. In order to achieve higher classification accuracy, we followed the method of [41]: Gray-level co-occurrence matrix (GLCM) texture features [42,43] and spectral indices were produced in order to serve as collective variable predictors for the classification algorithm. The texture characterizes the variance of the pixel DN value over space, so it needs to be measured in a multiple pixel neighborhood. Within this neighborhood of pixels can be found the following three elements: tonal (DN) difference between pixels, the distance over which this difference is measured, and directionality [42]. These neighboring pixels are considered a window or kernel, and usually have a square area with an odd number of pixels for practical reasons. It is important to define the window size, because the larger window size can include edges or patches with different textures; this is particularly applicable for larger window sizes. For the first time in 1973, Haralick et al. [42] proposed GLCM textures, which are co-occurring or second-order texture measures. The calculation is based on tonal (DN) differences in a spatially defined relationship between pairs of pixels, taking into account all pixel pairs within the neighborhood. Hall-Beyer explains that second-order measurements can distinguish two pixels wide vertical stripes from one pixel wide stripes, given uniform DN values in each stripe; first-order texture measurements are not able to perform this [42]. The GLCM can account for all three elements of texture and that is one of its advantages. GLCM can be calculated while using single input layer and defined window size (i.e., $7 \times 7$), selected by the user, and can deliver to one or more output layers based on the selected measurements (i.e., variance, homogeneity, entropy, etc.). Based on the empirical result, a window size of $7 \times 7$ yielded a better result for generation of GLCM textures and the following textures features were generated: Variance, Inverse Difference Moment, which measures the homogeneity, Contrast, Dissimilarity, Entropy, Correlation, and Angular Second Moment. which

measures the number of repeated pairs [42]. The GLCM of band 3 and Band 4 of Landsat 8 were used for 2019 land cover classification, and GLCM band 4 of Landsat 5 TM was used for 1992 to generate the texture features. The spectral indices used include: Modified Normalized Difference Water Index (MNDWI) [44], Enhanced Vegetation Index (EVI) [45], Normalized Different Moisture Index (NDMI) [46], Green Optimized Soil Adjusted Vegetation Index (GOSAVI) [47], Built-up Area Extraction Index (BAEI) [48], and the Normalized Difference Bareness Index (NDBai) [49].

The Smile RFC method was applied using 200 decision trees and eight variables per split, which accounts for two-third of all variables. The number of input variables for both years have been filtered according to the variable importance function of the RFC and only the variables that contributed most have been selected in order to produce the final study area land cover for the list of input variables.

Training data were collected from annual land cover data by ESA [50]. Stratified random sampling techniques were used to collect 500 points per class with a total 10,000 points. The overall classification accuracy reached over 80% for all time steps.

## 4. Results

### 4.1. Change in Temperature and Precipitation

In the KRB, two weather stations with more than 20 years of data are available, North Salang and Taliqan. Historical data are not available for the Kunduz meteorological station (see Figure 2). North Salang is located in the upstream, in a very high altitude with high precipitation and low temperature; Taliqan lies in the lowland area near of Kunduz.

Figure 5 shows a strong and statistically significant increase in the mean annual temperature within the KRB since the 1960s by 1.45 °C (see Table 2). All temperatures increase; however, the increase of the winter temperature is less and not statistically significant. Precipitation shows a very strong and significant trend by −35.02% (−412.56 mm) (see Figures 6 and 7).

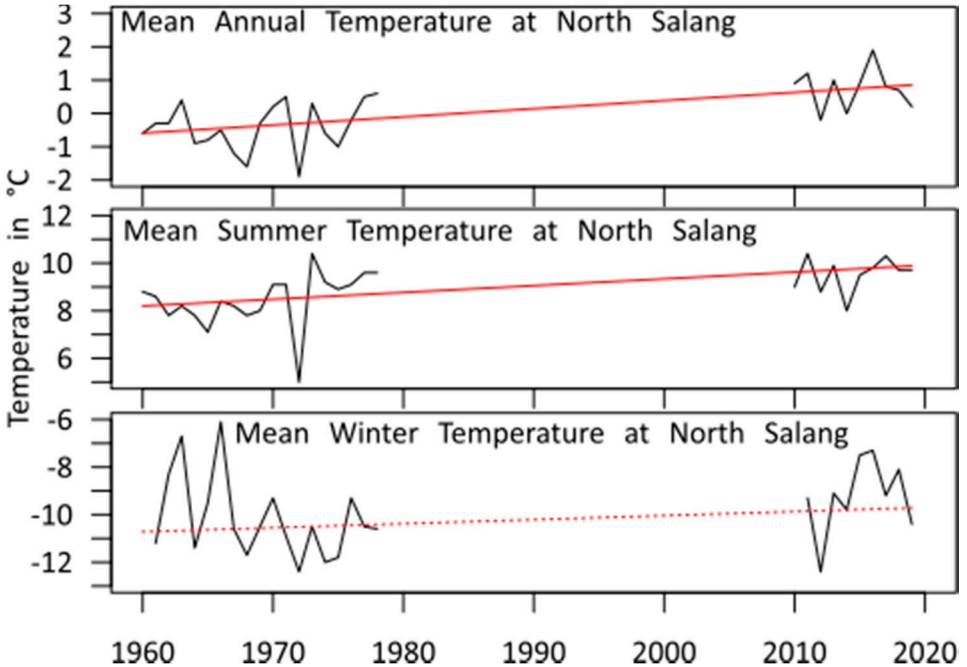

**Figure 5.** Mean annual, summer (J,J,A) and winter (D,J,F) temperature at station North Salang. Significant trends ($\alpha = 0.05$) are depicted as solid red line.

**Table 2.** Trends in temperature and precipitation for the stations North Salang and Taliqan. Statistically significant trends are bold (all but winter of North Salang).

| | Trend Mean Annual Temperature | Trend Mean Annual Spring Temperature (MAM) | Trend Mean Annual Summer Temperature (JJA) 1969–2019 | Trend Mean Annual Autumn Temperature (SON) 1969–2019 | Trend Mean Annual Winter Temperature (DJF) | Trend Precipitation 1960–2019 |
|---|---|---|---|---|---|---|
| North Salang | 1960–2019: **+1.45 °C** | 1960–2019: **+1.66 °C** | 1960–2019: **+1.69 °C** | 1960–2019: **+1.8 °C** | 1961–2019: +1 °C | −412.56 mm (−35.02%) |
| Talaqin | 1969–2019: **+2.73 °C** | 1969–2019: **+2.56 °C** | 1969–2019: **+2.87 °C** | 1969–2019: **+2.0 °C** | 1970–2019: **+3.68 °C** | −26.03 mm (−57.73%) |

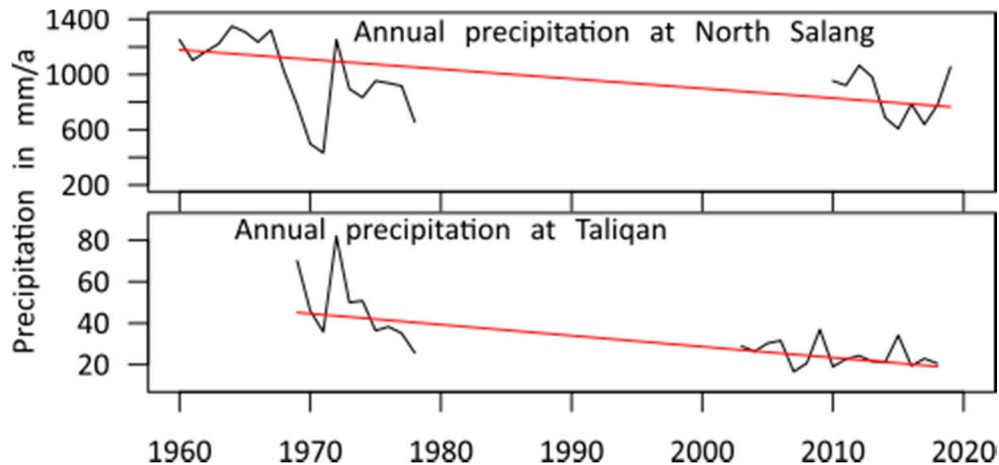

**Figure 6.** Mean annual precipitation at stations North Salang and Taliqan. All trends are significant t ($\alpha = 0.05$) and depicted as solid red line. Please note that only 18 years of data are available for Taliqan.

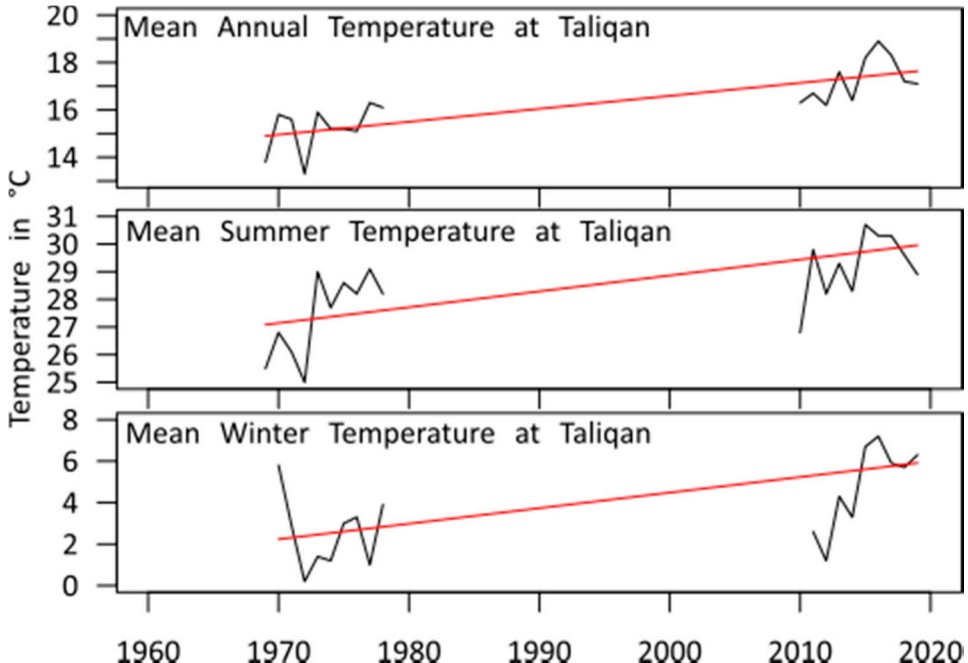

**Figure 7.** Mean annual, summer (J,J,A) and winter (D,J,F) temperature at station Taliqan. All of the trends are significant ($\alpha = 0.05$) and depicted as solid red line. Please note that only 18 years of data are available for Taliqan.

For the Taliqan station in the lowland, where only 18 years are available, all of the trends are significant and extreme. Mean annual temperature increased according to the data for the period from 1969 to 2019 by 2.73 °C and summer (+2.87 °C) and winter (+3.68 °C) temperature even more.

Precipitation decreased in the same period by 57.72% (−26.03 mm). These trends have to be interpreted with caution due to the limited number of years available (see Section 5.1).

*4.2. Changes in Discharge*

For this study, data from four gauging stations with at least 20 years of data in the KRB have been analyzed. The highland Doab station (see Figure 8, Table 3) shows a strong and significant increase in the mean and minimum annual streamflow, with over 100%, whereas the maximum flow is still strong, but due to the limited data not significant.

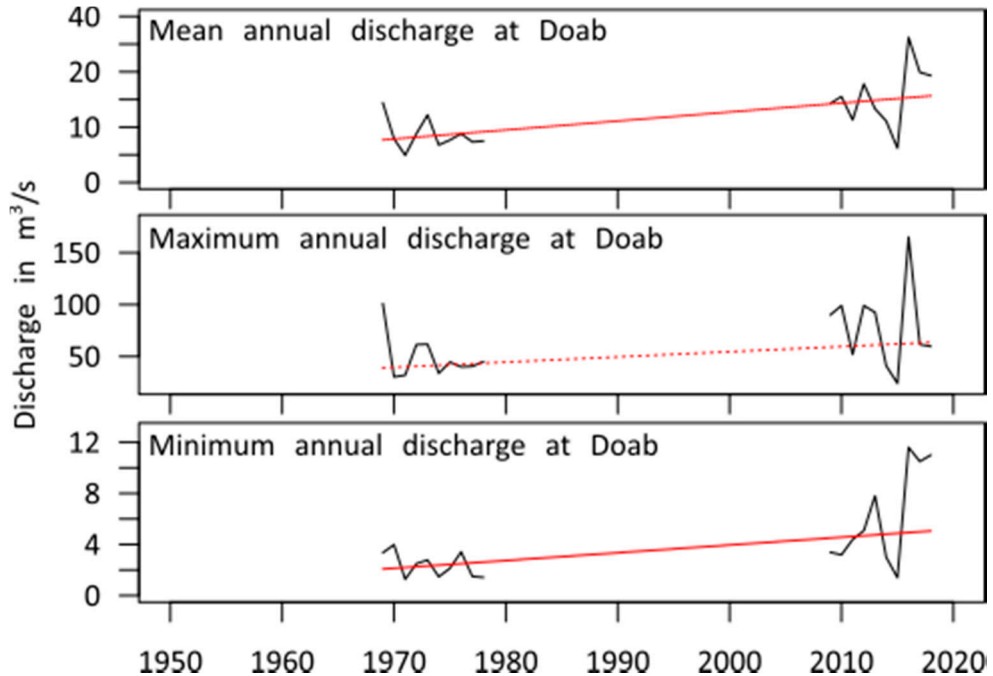

**Figure 8.** Mean, maximum, and minimum annual discharge for Doab gauging station. Significant trends (α = 0.05) are depicted as solid red line.

**Table 3.** Trends in mean, maximum, and minimum annual discharge for the gauging stations Doab, Pul-i-Khumri, Chahar Dara, and Kulokh Tepa. Statistically significant trends are bold (α = 0.05).

| Gauging Station | Trend Mean Annual Discharge | Trend Maximum Annual Discharge | Trend Minimum Annual Discharge |
|---|---|---|---|
| Doab | +7.95 m³/s (+103.12%) | +24.5 (+62.74%) | +2.98 m³/s (+143.27%) |
| Puli-Khumri | +5.38 m³/s (+7.86%) | −82.46 m³/s (−23.45%) | +11.39 m³/s (+53.34%) |
| Chahar Dara | −9.57 m³/s (−18.40%) | −125.53 m³/s (−43.05%) | −5.98 m³/s (−46.36%) |
| Kulokh Tepa | −27.46 (−25.30%) (significant at α = 0.1) | −334.61 (−58.51%) | −15.47 (−66.20%) |

The Puli-Khumri station, (Figure 9, Table 3) further downstream in the lowland of the KRB, shows inhomogeneous trends with an again strong and significant increase in minimum flow with over 50% decrease, whereas the maximum annual discharge is significantly decreasing by over 20% and the mean annual flow is consequently levelled out without a significant trend.

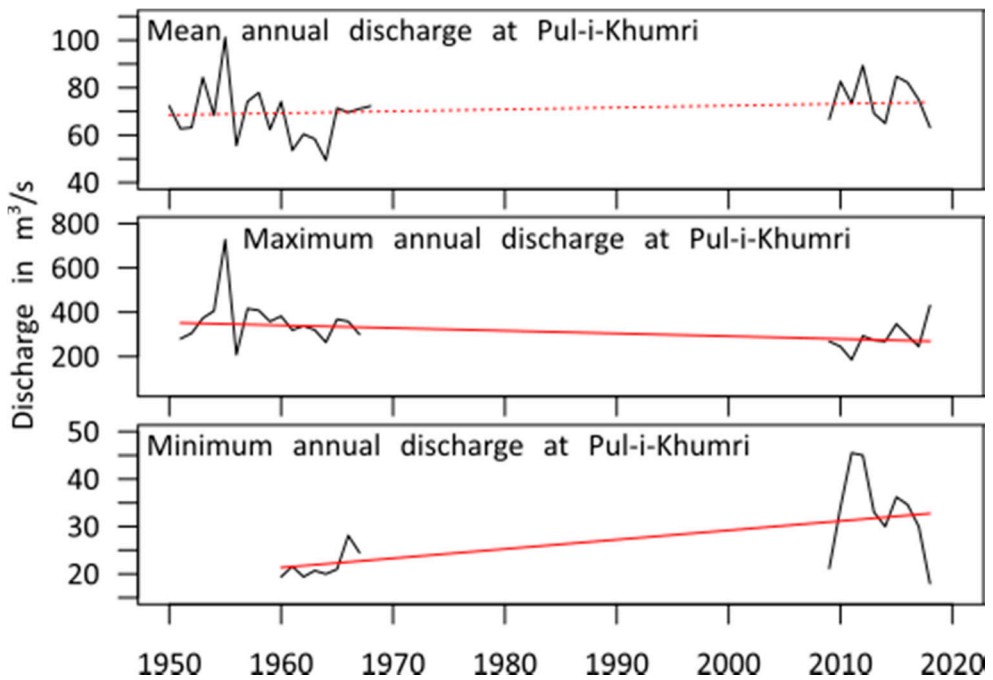

**Figure 9.** Mean, maximum, and minimum annual discharge for Doab gauging station. Significant trends ($\alpha = 0.05$) are depicted as solid red line.

The Chahar Dara gauging station (Figure 10, Table 3) further downstream shows strong decreasing trends throughout the year; however, only for the maximum flow this decrease is significant with over 40% reduction.

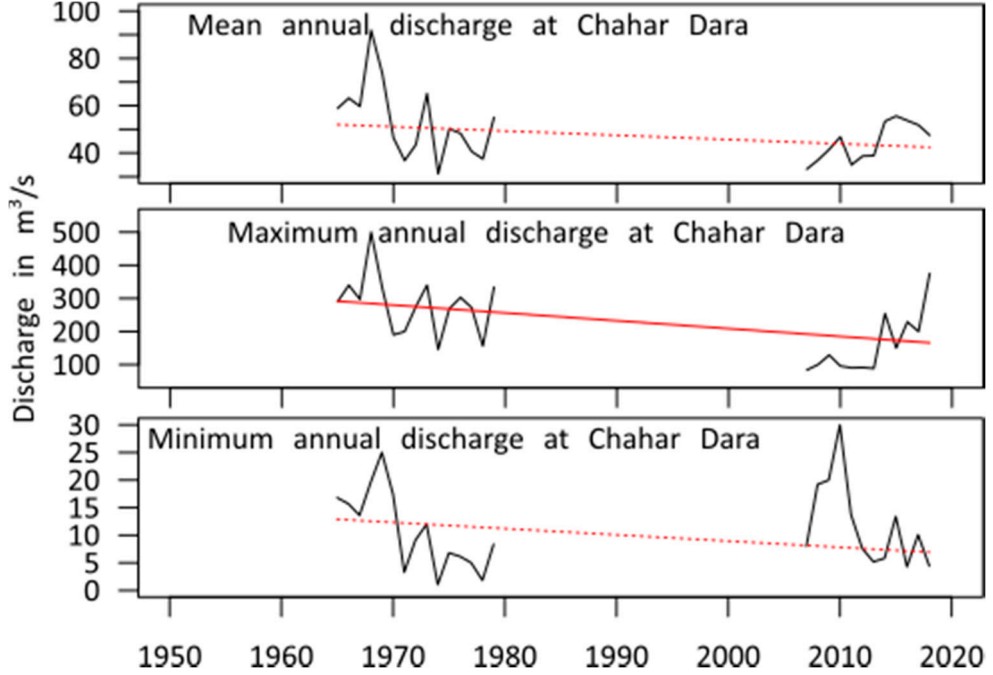

**Figure 10.** Mean, maximum, and minimum annual discharge for Chahar Dara gauging station. Significant trends ($\alpha = 0.05$) are depicted as solid red line.

The Kulokh Tepa station (Figure 11, Table 3) at the confluence of Kunduz the Amu Darya River shows similar decreasing patterns with a significant decrease in the maximum flow by almost 60% and a slightly less significant decrease ($\alpha = 0.1$) for the mean annual discharge by around 25%.

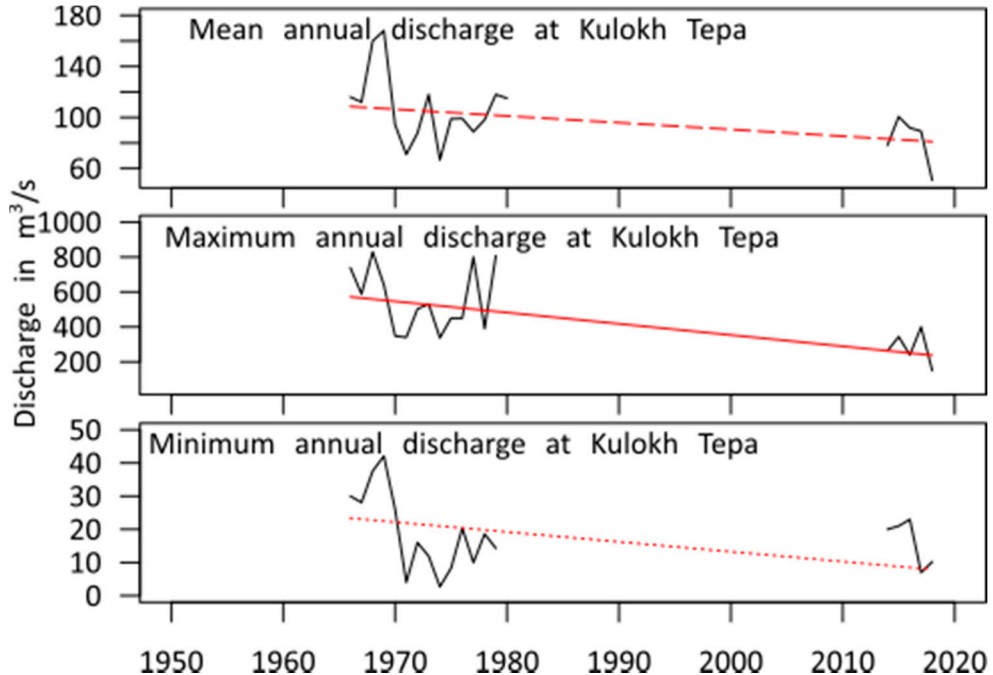

**Figure 11.** Mean, maximum, and minimum annual discharge for the Kulokh-Tepa gauging station. The significant trend with α = 0.1 is depicted as dashed, the significant trend with α = 0.05 is depicted as solid red line.

*4.3. Change in Landcover*

LULC trends in the KRB are assessed by comparing changes between the years 1992 and 2019 (Figure 12). Figure 13 shows the areal changes of the ten defined LULC types. Since 1992, irrigated agriculture, forest/trees, shrubland, urban coverage, as well as barren land and water surfaces, have increased substantially. At the same time, rainfed agriculture, grasslands, and snow/glacier coverage drastically decreased. Table 4 shows landcover classification area in Km² and the change in landcover percentage between 1992 and 2019.

**Table 4.** Comparison of 1992 and 2016 Landcover areas [23,28].

| Class Name | Landcover Area km²(1992) | Landcover Area km² (2019) | Change in % |
|---|---|---|---|
| Rainfed agriculture | 6382 | 4461 | −30.1 |
| Irrigated agriculture | 2064 | 2377 | +15.2 |
| Mosaic Vegetation | 12,847 | 12,488 | −2.8 |
| Forest, tree | 464 | 973 | +109.7 |
| Shrubland | 1859 | 3602 | +93.8 |
| Grassland/Rangeland | 9942 | 7361 | −26 |
| Urban | 266 | 548 | +106 |
| Bare land | 4964 | 5877 | +18.4 |
| Water | 174 | 249 | +43.1 |
| Snow/Glacier | 994 | 668 | −32.8 |

Forest/tree area also includes fruit trees and the doubling of this coverage can be explained by a massive expansion of fruit tree plantations, such as almond and pistachio trees. Grassland was mainly degraded to barren land or shrub land. There is also a shift from rainfed to irrigated agriculture, even though the decrease in rainfed agriculture cannot fully be explained by this shift. Large areas of rainfed agriculture seemed to shift into shrublands and barren land.

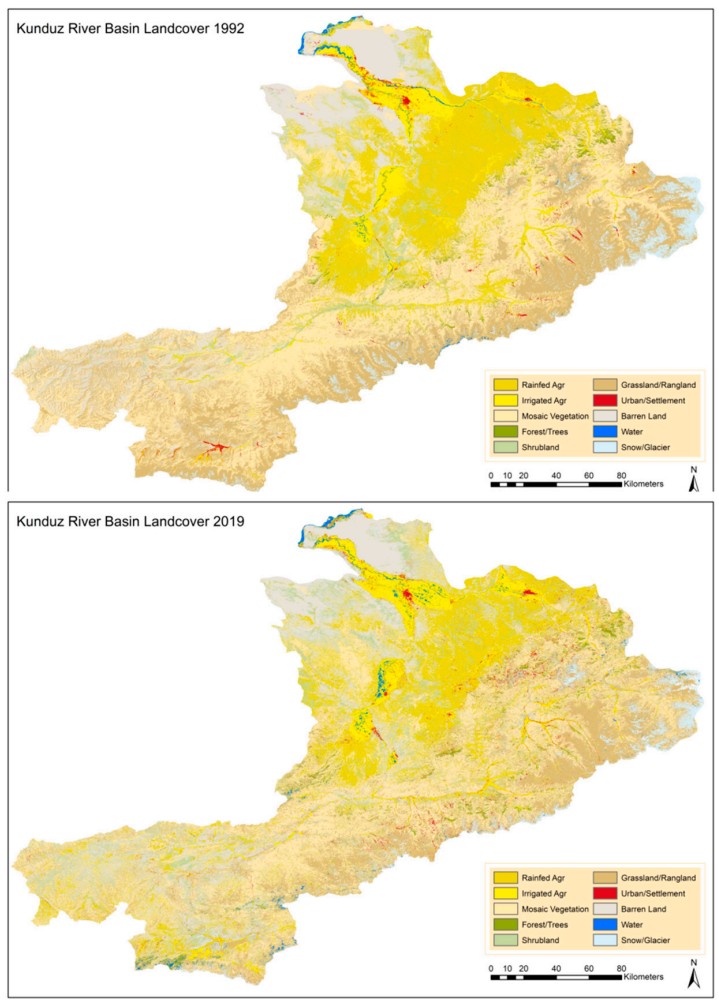

**Figure 12.** Land cover maps of 1992 and 2019 of the Kunduz River Basin derived from Landsat data.

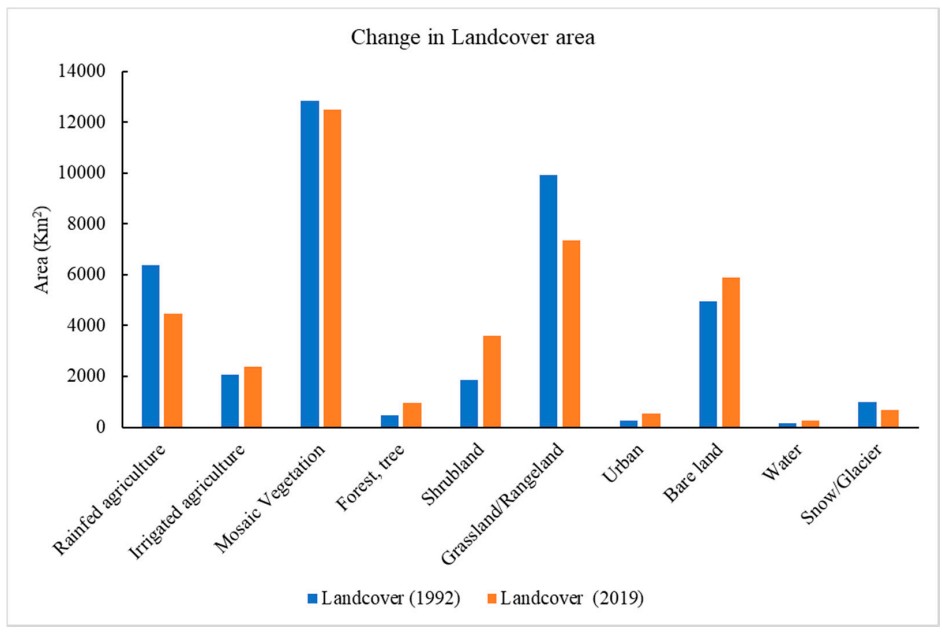

**Figure 13.** Changes in land use and land cover between 1992 and 2019 in the Kunduz River Basin.

## 5. Discussion

### 5.1. Constraints Due to Limited Data Availability

The availability of data is very limited in the study region as well as for the whole of Afghanistan for many reasons. The station density of meteorological as well as river gauging stations has always been low, due to the low population density, underdevelopment, and the relatively low influence of central government in many regions of the country. Characteristic for Afghanistan are, in addition the long periods of conflict and foreign rule, which hindered sustained observations or fragmented them. For example, weather observations from during the Soviet occupation, which still have taken place according to local knowledge, are not currently available. The lack of data also substantially reduces the quality of climate reanalysis in the region. Comparisons of observations with reanalysis for the available stations in the KRB showed the same results as for Aich et al. 2017 [9], which found that, for central Afghanistan, monthly precipitation in reanalysis deviated by up to 30% from the observations. For this reason, only observed station data are used for this study. We selected all of meteorological and river stations with at least 20 years of observations, since the IPCC AR5 used the period from 1986 to 2005 as modern baseline and deemed 20 years to be long enough to average over natural variations [51]. This filtering limited the time series for analysis to only two meteorological and four river gauging stations in KRB. Another constraint is the long gaps within the time series, which fragment the time series in two parts and make a continuous trend analysis impossible. The authors decided to use the data, despite these strong constraints, since it is still the currently best available data, which, in summary, still allow careful interpretation. The uncertainty of the temperature trend at the station North Salang is acceptable, since almost 30 years of data (29) are available, the value that the World Meteorological Organization (WMO) recommends for climate studies [52]. For precipitation, the uncertainty is slightly higher due to the strong interannual variability and long period of missing data. For Taliqan, the uncertainties are markedly higher, since only 18 years of data are available. Still, the temperature measurements give plausible results, even though the absolute numbers should be interpreted with caution. This holds even more for the extreme precipitation, which might be only natural variability.

However, both meteorological stations show consistent trends, which also confirm the findings from other studies with strongly increasing temperature and a reduction of precipitation [9]. This gives some confidence when interpreting the results and this holds also for the river stations. However, the climatic trends have the expected impacts on the river discharge in the KRB, even though the absolute numbers can be doubted. Finally, the individual time series can be questioned due to the mentioned constraints, but, all together, they show a coherent picture of a strong warming trend and drier conditions, which are also reflected by the changes of LULC.

In order to improve the situation and make more data available, we urge data rescue initiatives, like idare (https://www.idare-portal.org), to include Afghanistan in their efforts and particularly the integration of existing data in archives of the former Soviet Union might be promising.

### 5.2. Climate Change Impacts

The results of the temperature and precipitation trend analysis are, in general, in line with former studies, like Aich et al. 2017 [9]. The extreme increase in temperature by significantly over 1 °C in the central highland and even over 2 °C in the lowland of the KRB. The temperature increase is more pronounced in summer, accompanied by a not less extreme decrease in precipitation by over 35%, respectively, 50% during the second half of the last century until now (see Table 2). As discussed in Section 5.1, uncertainties with regard to the magnitude of trends is large, particularly for precipitation; however, the direction of trend seems to be plausible and in line with observations from other countries in the region. Still, the general decrease is significant and has, similar to the strong temperature increase, a strong impact on the water resources.

River discharge results are more heterogeneous for different parts of the catchment. In the headwaters of the catchment (Doab station), the discharge is significantly increasing, which can be explained by the increase of glacier melt due to the higher temperatures. The LULC analysis shows an extreme reduction of 359 km$^2$ (−35%) of glaciered area between 1992 and 2019. With the accelerated warming trend, the melting of the glaciers is also expected to accelerate and, at a tipping point, the increase in discharge in these upstream catchments will stop and discharge abruptly be reduced. Studies in other catchments in the Hindukush area show exactly this behavior, with a current increase in discharge in the headwaters, but project a strong decrease on the long run [2,53]. The warming is, in general, altering the flow regime in the whole catchment, since the period of snowfall is reduced and precipitation, which is usually stored until spring as snow cover, feeds as direct runoff into the river systems.

In the Puli-Khumri station, which is already in the lowland of the catchment, the decrease of precipitation already leads to a decrease in maximum annual discharge, even though this is leveled out overall by the additional discharge through the glacier melt. For the other stations further downstream, the increase in evapotranspiration that is caused by the increased temperature and the strong reduction of precipitation leads to strong decrease in streamflow. This holds for both maximum and minimum discharge, but it is most pronounced during the summer discharge peak.

This interpretation of the results is also supported by the trends in change of landcover, which show a general tendency to drier conditions and a significant increase in human activities. The reduced rainfall and increased evaporation caused a reduction of grassland and an increase of barren land. Parts of rainfed agriculture have been turned into irrigated agriculture, but large parts have also been abandoned and turned into shrubland and barren land. A plausible explanation for this observation might be the drier conditions, which do not allow rainfed agriculture in many parts of the basin anymore. On the other side increased forest and tree cover, which can be explained by the substantial increase of fruit tree cultivation, which are more resilient to the drier conditions in the catchment. In addition, urban settlements increased strongly, which likely puts even more pressure on the land and available water resources.

## 6. Conclusions and Recommendations

The study results indicate that, since the 1960s, the annual average temperature in the KRB has been increasing, while precipitation and river discharge have been decreasing, with the exception of glacier-fed headwaters. The increase in the discharge in the upper catchment will continue until the small glaciers that still exist are melted and then a dramatic decrease in summer discharge where it is most needed for irrigation can be expected for the whole catchment, similar to other catchments [54]. In addition, there has been a drastic and significant change in landcover since 1992, most likely due to climate change impacts as well as environmental degradation and human impact. This leads to more direct run-off of precipitation which increases the risk of floods. In combination, these processes negatively impact the livelihoods and wellbeing of its communities.

About 1.9 million people live in the KRB and their livelihoods mostly rely on agriculture. Climate change impacts therefore affect food security, particularly of those depending on the household farming. Decreasing precipitation results in a depletion of water resources, in some cases leading to water scarcity. In addition, the combination of climate change impacts and strong pressure on the land use during the long period of war and conflict in the country has led to a degradation of vegetation cover in the KRB. Afghanistan is traditionally an agrarian country, with 22% of the national GDP produced in this sector. Approximately 79% of the population is engaged in farming. Agriculture is an important source of livelihood and local economy rely on that [55,56]. Agriculture and farmers are more affected by the impacts of climate change in Afghanistan [57]. The main obstacles are war and conflict in the country and a lack of effective investment and management in agriculture and irrigation sectors. Additionally, land use and land cover change due to socio-economic changes through political and economic transformation and climate change impacts is a critical issue in Afghanistan and a number of

studies have been conducted on LULC in Afghanistan [58–61]. A LULC study undertaken by FAO as compared LULC for Afghanistan between1993 and 2016 showed changes in the KRB land cover that are in line with the results of this study [19,23].

In turn, this affects the capacity of people and the environment to adapt to climate change. The strong warming trend in winter and spring lead to an earlier snow melt, which again increases the risk of flash flooding. However, there are also positive signals visible. There is a strong increase of fruit trees, which are more resilient to harsher climatic conditions and may even locally have the positive effect the microclimate. In addition, irrigated agriculture has also increased. Both of the signals show that farmers adapt automatically autonomously to the changing conditions.

In addition, the study shows that the annual discharge of the KRB is sufficient for developing the watershed if the water resources are managed in an integrated and sustainable way. The downstream part of the KRB covers a wide area with large agriculture potential, for example by multiple cropping through irrigation. At the same time, the downstream part of the KRB is very vulnerable to flash floods and droughts, which affect the livelihood and socio-economy of the community living within the watershed deeply. Therefore, integrated water resources management is key for the agricultural development, livelihoods, and local economy. Measures, like reforestation, could reduce the risk of flash floods and droughts. Other measures, which have proven their effectiveness for many catchments in a developing context, could include guidelines on best practices, the establishment of a river basin council, and adapted community-based participation approaches. Using approaches that directly involve the communities in management and decision-making processes, these collectively can improve the socio-economy and livelihoods of the people within the KRB. However, a comprehensive IWRM strategy is still missing for Afghanistan and particularly the KRB. Therefore, we hope that the results of this study contribute to informing sustainable water resources development and watershed management. In conclusion, this study argues for establishing an Integrated Water Resources Management Plan for the KRB to trigger sustainable development [62].

**Author Contributions:** Conceptualization, N.A.A. and V.A.; Methodology, N.A.A., S.S. and V.A.; Writing—Original Draft Preparation, N.A.A.; Writing—Review & Editing, N.A.A., S.S. and V.A.; Visualization: N.A.A., S.S. and V.A., Funding: N.A.A. All authors have read and agreed to the published version of the manuscript.

**Funding:** This research received no external funding.

**Acknowledgments:** We thank Susan Cuddy from CSIRO Land and Water branch for her revision of the paper, susan.cuddy@csiro.au.

**Conflicts of Interest:** The authors declare no conflict of interest.

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
