# Peer review of "Impacts of Climate Change on the Water Resources of the Kunduz River Basin, Afghanistan"

_climate, doi:10.3390/cli8100102_

Round 1

Reviewer 1 Report

Dear authors,

this article version is much better than the previous one. Congratulations!

Reviewer 2 Report

The authors answered my questions and comments sufficiently. The paper could benefit from further text editing, but overall it is clear and readable.

Reviewer 3 Report

The authors have addressed to the comments. The quality of the manuscript is acceptable now.

This manuscript is a resubmission of an earlier submission. The following is a list of the peer review reports and author responses from that submission.

Round 1

Reviewer 1 Report

General comment:

The sources are missing under each picture or table!!!!!

Specific comments:

As the tile of the article is "Impacts of Climate Change on the Water Resources.... I would expect some synthesis made on the individual analytical results. This is very essential issue, otherwise it is not very interesting for readers and one does not know what actually you want say by this article.

In the results part, you are presenting just the individual analysis of changes in temperature and precipitaion,  changes in discharge, change in landcover. But what does this mean for water resources as a whole? Again, the synthesis is missing.

Furthermore, in conclusions I would like to suggest to add 1-2 paragraphs about your proposals for integrated water resources management. In the beginning you have mentioned that a lot of communities live there. So what do you propose?. How can be your results utilised for water resources management? Otherwise it does not make any sense to have just individual analysis.

Reviewer 2 Report

Dear Editor,

In ‘Impacts of Climate Change on the Water Resources of the Kunduz River Basin, Afghanistan’, the authors describe an analysis of two precipitation time series and four river flow time series combined with an analysis of land cover change using satellite images to understand the impact of climate change on this basin.

The HKH region is an important climate change hotspot and studies from Afghanistan, or the western part of the HKH in general are sparse so any analysis would be a useful addition to the literature. I read the manuscript with interest and can see it can add valuable insights, but I believe it would require further modifications, especially regarding the land cover contribution.

In the abstract the land cover analysis is introduced as “A comparison of landcover data … from 1992 and 2019 shows significant changes … which are used to interpret the trend analysis.” It is given equal weight to the timeseries analysis in methods and results. I can see the validity of performing such an analysis but I have difficulty judging the results; i. from methods section I understand that different bands are used from two different types of images; how can the authors be sure that the relatively modest changes (in area, but also in the type of land cover; shrubs or rainfed agriculture or grassland might not be so easily distinguishable, under varying weather conditions; and I only two maps are shown, which makes it hard to interpret the differences. On a more fundamental level, I do not think changes in land cover can easily be attributed to climate change in this basin; there have been large geopolitical and socio-economic changes over the time period considered, as the authors also mention, which have likely impacted landcover as well. These impacts cannot be separated from climate change effects with the approach used (if combined with a model, one might be able to attribute). Grassland is both converted to shrubland and forests (or tree plantations) and to bare (barren) land – the former seems to relate to drought but not the latter. What to make of that? And what could be the impact of weather during the particular months? If one summer was much drier than the other, could that have affected the images and these results?

Therefore, I don’t think the authors should make strong statements such as: L378 ”drastic and significant change in landcover since 1992, most likely due to climate change impacts”, and repeated in the abstract. This cannot be concluded from the limited land cover assessment performed.

If the authors would be able to provide evidence of land use change from other sources, this might help understand what is shown. A few times ‘local farmers’ are mentioned (e.g. L381) but nowhere is explained how these were contacted and what information they might be able to provide. It would be interesting though to see more detail. I am, e.g., also surprised to read that fruit trees are planted because they are more resilient to climatic conditions (L393). Mostly the literature highlights the risk of climate change to tree crops because of the large investment made, so this would be an interesting small point of discussion.

Please add a reference to the suggestion that tree crops might locally slow warning, and rephrase (I guess the authors mean they might influence the microclimate, but that is different from slowing the warming).

My second concern is with regards to the presentation of the trends based on few years with long gaps in between (so not the trends itself). I appreciate the effort is being made, and the methods (Mann Kendall and Theil-Sen approaches seem appropriate. But fact remains that there are very little data points, especially for the second period, and that drawing a trendline through open space just looks odd. Could the authors discuss to what extent decadal variability can be ruled out? Why plot a non-significant trends – what do you want to suggest with such a dotted line? (suggestion to leave out). Visually, trend lines covering a huge data gap makes one question; I wonder whether a simple t-test of the means between the two periods would not be better. I expect results to be the more or less the same and whether differences represent a ‘trend’ can then be discussed and/or left to the reader. If the authors would like to stick to the MK and TS approaches, I think it would help if they are more explicit that these are suitable to be used with this kind of discontinued data.

Regarding the periods presented; several times it seems more interesting what is happening in the periods not shows; e.g. for figure 5 the annual trend is stronger than either summer or winter which suggests something must be changing in autumn or spring. Can this be added, maybe in a table or SI.

At the same time, there is duplication of results. Figure 5, the text and table two all repeat the same results. Please combine (e.g. plot the figures in table 2 in figure 5?, and only summarize in the text, not repeat values).

Please avoid description such as L230 ‘extraordinary extreme’ – keep it neutral, judgement is up to the reader.

Other comments:

Sections 2.2 and 2.3 seem to contain results; if this is a paper on climate using two weather stations and four available flow gauges, I expect the numbers to come back in the results, e.g. as a first section, not in the methods. Advice to shorten this and to merge it with 2.1.

Figure 1 is quite different from figure 32 in layout and not very clear. Use layout from figure 2 for figure 1 as well?

L200 what kind of classification? Later on I see you used training data so I guess a classified one?

L201 “median function of GEE has been used, which calculate the median of each pixel for two points” Did you calculate the median of two points…?

L204-205. Please write out ref 42 and explain these sentences. This section is not easily readable and understandable for the readers of Climate

L375-376 “ The increase in the discharge … will continue until the small glaciers that still exist are melted and then a dramatic decrease in summer, discharge where is most needed for irrigation, can be expected for the whole catchment.” This is not your finding, but an extrapolation, i.e. an assumption. Please either cite literature supporting this assumption or leave out.

More in general: the authors highlight several times the scarcity of data and difficult conditions for collecting data. Please limit this to intro, addressing the reason/scope, and methods, with possibly a sentence or two in the discussion (but not a whole paragraph repeating again).

Another proofread would further improve the manuscript.

Reviewer 3 Report

I appreciate the Editor to give me a chance to review an interesting and valuable paper. I found some merits in the both methodology and results. In my opinion, this paper has a good potential to be published in the journal. However, I have also some concerns on the different parts of the manuscript. If the author(s) address carefully to the comments, I’ll recommend publication of the manuscript in the journal:

  • In the last paragraph of the Introduction, the authors should clearly mention the weakness point of former works (identification of the gaps) and describe the novelties of the current investigation to justify us the paper deserves to be published in this journal.
  • I strongly recommend the authors to use trend-free pre-whitening (TFPW) method to eliminate serial correlation of time series. This improve the results and leads to more reliable results.
  • In the Tables, highlight values that are more important and discuss them for better understanding readers.
  • How can extend the results in other regions with similar/different climates?
  • At the end of the manuscript, explain the implications and future works considering the outputs of current study.
  • The newest paper cited in the reference list has been published in 2018. In addition, many of the references are national papers or websites/reports. The authors should add some most recent references (2019 and 2020) and compare their results with them to indicate the advantages of the current study.
  • The quality of the language needs to improve by a native English speaker for grammatically style and word use.